# Standardising Training of Nurses in an Evidence-Based Psychosocial Intervention for Perinatal Depression: Randomized Trial of Electronic vs. Face-to-Face Training in China

**DOI:** 10.3390/ijerph19074094

**Published:** 2022-03-30

**Authors:** Anum Nisar, Juan Yin, Yiping Nan, Huanyuan Luo, Dongfang Han, Lei Yang, Jiaying Li, Duolao Wang, Atif Rahman, Xiaomei Li

**Affiliations:** 1Health Science Centre, Xi’an Jiaotong University, Xi’an 710049, China; nisaranum@stu.xjtu.edu.cn (A.N.); lanshui.123@stu.xjtu.edu.cn (Y.N.); dongfangh0928@xjtufh.edu.cn (D.H.); yanglei678@xjtu.edu.cn (L.Y.); janice123@stu.xjtu.edu.cn (J.L.); 2School of Nursing, Dalian University, Dalian 116622, China; yinjuan@xjtu.edu.cn; 3Global Health Trials Unit, Liverpool School of Tropical Medicine, Liverpool L3 5QA, UK; 238551@lstmed.ac.uk (H.L.); duolao.wang@lstmed.ac.uk (D.W.); 4Department of Primary Care and Mental Health, University of Liverpool, Liverpool L69 3BX, UK; atif.rahman@liverpool.ac.uk

**Keywords:** perinatal depression, technology, training, psychosocial intervention, Thinking Healthy Programme

## Abstract

Background: Rates of perinatal depression in China are high. The Thinking Healthy Programme is a WHO-endorsed, evidence-based psychosocial intervention for perinatal depression, requiring five days of face-to-face training by a specialist trainer. Given the paucity of specialist trainers and logistical challenges, standardized training of large numbers of nurses is a major challenge for scaling up. We developed an electronic training programme (e-training) which eliminates the need for specialist-led, face-to-face training. The aim of this study was to evaluate the effectiveness of the e-training compared to conventional face-to-face training in nursing students. Methods: A single blind, non-inferiority, randomized controlled trial was conducted. One hundred nursing students from two nursing schools were randomly assigned to either e-training or conventional face-to-face training. Results: E-training was not inferior to specialist-led face-to-face training immediately post-training [mean ENhancing Assessment of Common Therapeutic factors (ENACT) score (M) 45.73, standard deviation (SD) 4.03 vs. M 47.08, SD 4.53; mean difference (MD) −1.35, 95% CI; (−3.17, 0.46), *p* = 0.14]. There was no difference in ENACT scores at three months [M = 42.16, SD 4.85 vs. M = 42.65, SD 4.65; MD = −0.481, 95% CI; (−2.35, 1.39), *p* = 0.61]. Conclusions: E-training is a promising tool with comparative effectiveness to specialist-led face-to-face training. E-training can be used for training of non-specialists for evidence-based psychosocial interventions at scale and utilized where there is a shortage of specialist trainers, but practice under supervision is necessary to maintain competence. However, continued practice under supervision may be necessary to maintain competence.

## 1. Introduction

Globally, perinatal depression is a common mental disorder during pregnancy and after childbirth with prevalence rates of 7% to 15% in high-income countries and 19% to 25% in low and middle-income countries (LMICs) [1]. The prevalence in China ranges from 15–20% [2], and appears to have sharply increased during the COVID-19 pandemic [3]. Perinatal depression causes suffering not only to the mother but is associated with preterm birth, low birth weight, poor breastfeeding practices, high rates of diarrheal episodes, and poor cognitive development in infants [4]. Despite such public health implications, perinatal mental health services in China, especially in the underdeveloped areas, are inadequate and face challenges, such as lack of trained specialists [5,6,7]. China’s National Mental Health Working Plan (2015–2020) and the recently implemented national mental health laws in 2015 are helping to develop capacity in acute and community-based care for common mental disorders, regulating the diagnosis and treatment of mental disorders, and establishing health institutes and institutional capacity for health professionals in the diagnosis and treatment of mental disorders [8]. The integration of mental health care with community-based services for early detection, treatment, and rehabilitation is also prominent in these plans. These national reforms aim to develop integrated models of management of mental disorders, incorporating primary health, judicial administration, and public security sectors of government. Building workforce capacity is a focus of national mental health reforms for both acute and community-based care. The potential role of nurses in meeting this demand for a mental health workforce and tailoring their training needs accordingly has been highlighted [9].

Over the last decade, the global evidence for effectiveness of psychosocial interventions delivered by non-specialist health workers (such as community health workers, nurses, health visitors, and midwives) has been building [10]. The technology-assisted delivery of psychological interventions, such as on-line and mobile applications (apps), has been demonstrated successfully, specifically in high income countries (HICs) [11]. Lower- and middle-income countries are also implementing digital means of training to bridge the higher treatment gap [12]. Perinatal depression, due to its impact on women and children, has been prioritized [13]. The World Health Organization (WHO), following a systematic review of available evidence for such interventions, recommended the Thinking Healthy Programme (THP) as the treatment of choice for women requiring psychosocial management for their perinatal depression in primary and secondary care settings [14]. Thinking Healthy Programme (THP) is an evidence-based psychosocial intervention that can be delivered by non-mental health specialists [15]. THP has been translated and adapted for the Chinese population according to the Bernal framework of translation and adaptation [16]. Moreover, cultural relevance and acceptability of THP to be delivered to the Chinese perinatally depressed mothers by non-specialists has been established [16].

A key feature of the Thinking Healthy Programme is that non-specialists can be quickly trained to deliver the programme under specialist supervision, and the intervention can be integrated into primary and secondary health care [17]. Thus, it is ideally suited to regions where there are not enough trained specialists for intervention delivery. THP, delivered by community health workers, was tested in a large community-based randomized controlled trial in Pakistan, where it more than halved the prevalence of perinatal depression and significantly improved child health outcomes, such as diarrheal episodes and vaccination coverage [15]. THP is part of the WHO’s flagship mental health gap action programme (mhGAP) and has been translated into a number of languages and is being tested in many countries [18]. In a previous study, we adapted the Thinking Healthy Programme for China, the world’s most populous country, with its own distinct sociocultural and healthcare system. THP was found to be relevant and acceptable in this context [16].

A major hurdle in scaling-up the programme remains the shortage of specialists to train and supervise health workers. Although the training is brief (five days), the conventional method would still require thousands of specialist trainers to conduct face-to-face workshops in a scaled-up programme, which would be a challenge in a country as vast as China. In a similar context, in Pakistan, we developed a technology-assisted training programme for community health workers in a conflict-affected area of Pakistan and found it to be as effective as face-to-face training [19]. Moreover, numerous online-training programmes for mental health care professionals have seen satisfactory results in many parts of the world [20,21,22]. Literature also suggests that online-training methods can be an effective way to train behavioral health care professionals [23]. To address the scale-up challenge, we adapted this e-training for nurses in China. Our goal was to incorporate training of this evidence-based psychosocial intervention into the routine curriculum of nurses without the need for a specialist trainer.

The aim of this study was to compare the adapted electronic version of training delivered by a non-specialist trainer with conventional specialist-delivered face-to-face training in two nursing schools in Xi’an, China and to examine the competency achieved in delivering the interventions in both groups. This study evaluated the classroom component of the training only, as the students did not go on to deliver the intervention to patients.

## 2. Materials and Methods

Trial registration: The trial was registered at www.chictr.org (ChiCTR1900027389), accessed on 1 January 2022.

### 2.1. Study Design

A single blind, non-inferiority, individual randomized controlled trial design was employed, comparing the competence of nurses receiving electronic training (*e*-training) versus those receiving conventional face-to-face training. The non-inferiority design was chosen because a new method of training was being compared to an established standard method of training. The primary outcome of the study was the mean competence scores immediately post-training. Feedback from participants about training was also collected during and at the end of both the trainings.

### 2.2. Participant Flow and Numbers Analysed

One hundred students participated in the study. The students were equally randomized into intervention and control arms. All the participants completed the training and post-training assessment. At the primary endpoint (immediately after training), 96 out of 100 students (96%) completed the assessment. (Figure 1 given below).

### 2.3. Settings and Participants

The study was undertaken in Xi’an city in the Shaanxi province in northwestern China. Xi’an has a population of 8.8 million and is the most populous city of Northwest China. The birth rate is 12.6% and there were 118,000 newborns in 2017. There are 3.2 doctors and 4.3 nurses per 1000 residents, respectively. The hospital delivery rate in Xi’an is 99.9% [24]. The study was conducted in two large public nursing schools in Xi’an, the City College for Nursing and the Faculty for Nursing at the Xian Jiaotong University. Both institutions provide graduate level training to nurses. Participants were nursing students enrolled in the four-year nursing training programme.

#### Demographic Characteristics of Participants

The mean age of participants was 20 years. None had prior mental health work-experience. There were no significant differences in demographic variables measured in both arms (Table 1).

### 2.4. Sample Size, Sampling Method, and Inclusion Criteria

The primary outcome of the study was the mean competence scores immediately post-training, with a secondary time point of assessment three-months post-training. Based on our previous study in Pakistan [19], we defined non-inferiority as a difference of five points or less (corresponding to a 10% difference on the outcome measure score) in the mean competence score between the two groups measured with a standardized tool. A sample size of 100 students (50 students in each arm) provided 99% power to show non-inferiority of e-training to face-to-face training after accounting for an attrition rate of 15% at three months follow up to detect a five-point difference with a 0.05 one-sided alpha level.

From the list of students enrolled in the two colleges, all students were approached to take part in the study. The first 100 students who provided informed consent and whose schedules allowed them to take part in the study were recruited into the trial. Participants were included if they intended to complete their nursing training and had access to WeChat.

### 2.5. Randomization and Masking

The unit of randomization was the students. We randomly allocated the 100 students on a 1:1 ratio, stratified on the basis of nursing school (equal number of students from each nursing school). Randomization was conducted by an independent, off-site team member using computer software. Allocation concealment was ensured by keeping the random assignments in sequentially numbered, opaque, sealed envelopes at the off-site centre. Outcome assessors were blind to the allocation status.

### 2.6. Intervention Arm: E-Training in the Thinking Healthy Programme

A tablet-based, multimedia training programme was developed, derived from our previous work in Pakistan [19] and using the Chinese version of THP [16]. Following the methodology of our Pakistan study [19], the adapted Thinking Healthy training manual was converted into narrative scripts by the Mandarin-speaking authors (JY and XL). Training narratives incorporated all the core techniques and principles of the intervention, such as effective use of counselling skills, collaboration with the mothers’ families, guided discovery (approach to know the mothers’ mental health attitudes), and setting health-related tasks. Culturally appropriate real-life characters for the trainers and the trainees were developed. A Chinese artist converted the characters into “avatars” (graphic image of the characters), which were used to voice the narrative scripts. The prototype e-training was tested with users (THP trainers and nurses) during and after the development process. Feedback on all aspects of the system, including the user interface, was recorded and further refinements were carried out. Following development, five-days of e-training were incorporated into the curriculum of the student nurses. Training was delivered by demonstrating the videos developed followed by discussions and role plays to reinforce key messages and develop intervention-delivery skills. Role plays were structured and meant to practice the key skills in delivery of a session as well as management of challenging situations and methods of dealing with adverse events. After observing these, students were required to practice the skills acquired through role plays amongst themselves. Peer assessment of role plays, reflections on learning, sharing of relevant experiences, and discussing problem-solving strategies further reinforced the training. This entirely automated, classroom-based training was facilitated by their regular tutor.

### 2.7. Control Arm—Conventional Specialist-Delivered Training

The training to the control arm was delivered by a specialist trained by a WHO-approved Thinking Healthy master trainer. The conventional face-to-face class-room training was conducted using the translated and adapted paper versions of the THP manual, the training guide, and a set of power-point slides. The training was a five-day duration and consisted of lectures, group discussions, role plays, and feedback on the role plays by the trainer and students. The specialist trainer facilitated the entire training.

### 2.8. Measurements

Training outcomes generally include provider-level outcomes, including (a) competence (skill with which EBP techniques are delivered), (b) fidelity (long term sustainability of competence), (c) knowledge (understanding of principles and techniques), (d) satisfaction (acceptability of the training to providers), (e) skill acquisition and confidence (confidence in being able to employ techniques learned), and (f) adoption (use of intervention by a provider post-training) [25,26]. Implementation-level outcomes include training costs, such as financial resources and provider time. Service-level outcomes include symptom reduction, functional improvement, and client treatment satisfaction. Implementation and service outcomes were not in the scope of this study, which focused on provider-level outcomes (a to e).

### 2.9. Primary Outcome

Competence and fidelity in training immediately post-training and at three months post-training were measured by the ENhancing Assessment of Common Therapeutic factors (ENACT) rating scale, developed by Kohrt and colleagues [27]. ENACT is an 18-item scale to assess competence of non-specialists via role plays or observation of a therapy session. ENACT was developed using a rigorous methodology and has shown good psychometric properties [27]. Each item (also called a domain) is scored on a scale from 1 to 3, where 1 = needs improvement, 2 = partially done, and 3 = done well. A composite score can be computed by adding all the individual item scores. ENACT has been contextualized and implemented successfully in various countries [28,29,30]. We have used ENACT in a previous study in Pakistan to measure competence of community health workers in the Thinking Healthy Programme [19]. As in our previous study, we used an adapted version of ENACT (excluding items 17 and 18 as these were more systems-related concerning confidentiality and risk management). (The domains are summarized in the Appendix A). In line with our other studies using ENACT in similar trainings of non-specialist health workers [31], we used the cut-off score of 80% or more to indicate competency as recommended by the developers of the ENACT.

The competency assessments were conducted by independent assessors who rated videos of the role plays of students enacting a set of therapeutic encounters. The students were asked to undertake structured role plays which were devised so that they covered the key domains assessed by ENACT. These videos were taken immediately post-training and at three months post-training. Each recording was evaluated by two independent members of the research team who were blind to the allocation status of the participants with agreement required between both for a final score.

### 2.10. Secondary Outcomes

#### 2.10.1. Counsellor Self-Efficacy

The counsellor activity self-efficacy scales (CASES) [32] were used to assess the student’s self-efficacy in performing the intervention. CASES measure self-efficacy in three domains: performing helping skills, managing the counselling process, and handling challenging counselling situations. The measure consists of 41 items that are answered on a 10-point scale of confidence with a rating of 1 indicating no confidence at all to 9 indicating complete confidence in one’s ability to perform particular psychosocial skills and specific intervention-based tasks. The six subscale scores are then added to obtain an overall score ranging from 41 to 410. Higher total scores indicate greater levels of perceived self-efficacy, and lower total scores indicate lower levels of perceived self-efficacy.

#### 2.10.2. Attitudes and Beliefs

Items related to attitudes and beliefs were adapted from the perinatal depression monitor, a tool used in depression literacy surveys in Australia [33]. The questionnaire consisted of 12 items. Participants responded on a scale of 1 to 7.

#### 2.10.3. Satisfaction with Training

A specially developed, semi-structured questionnaire was used to obtain feedback on training from participants in both arms. It consisted of five items exploring their subjective experiences of the training, including (a) how much they had learned; (b) how useful they found the training; (c) if they would use the knowledge and skills learned in their future work; and (d) if they would recommend the training to others.

We also collected basic sociodemographic information, such as age, years of education, and previous training in mental health. We also assessed baseline knowledge of perinatal depression through a specially developed questionnaire that included 12 items (yes or no responses with a maximum score of 12) derived from their routine curriculum and covered areas such as presentation, prevalence, risk factors and impact of the condition.

### 2.11. Data Collection Procedures

All secondary outcomes were assessed using self-administered questionnaires. We used the WeChat platform to distribute the questionnaires to our participants. WeChat is the most commonly used social media platform in China with over one billion users. Every student has a WeChat account which is frequently used for educational activity and that they agreed to share with the research team for the purposes of assessment. The questionnaires were sent immediately after the training and three months post-training. Participants were sent two reminders if they did not respond.

### 2.12. Statistical Analysis

Baseline characteristics (age, gender, grade, work experience, and prior mental health training) by study arms (intervention and control) were summarised. All outcomes, including ENACT scores, perceived self-efficacy, attitudes, beliefs, and satisfaction with training by study arms at post-training and three months post-training were provided descriptively as mean and standard deviation.

Primary analysis was conducted using the generalised linear mixed models (GLMMs) with study arm (intervention or control), follow-up visit (post-training or three-months follow up), interaction of study arm, and follow-up visit as fixed effects, and subject as random effect (results given in Table 2). In addition, the ENACT score was converted into a binary variable using the cut-off score of 43 (i.e., total score 54 multiplied by 80%) to indicate whether a student had achieved the required level of competence, and it was analysed using GLMM.

Covariate adjusted analysis was conducted using GLMMs with study arm (intervention or control), follow-up time, interaction of study arm and follow-up time as fixed effects, subject as random effect, and baseline characteristics as covariates.

Missing outcome data were imputed using the last observation carried forward (LOCF) method. Summary statistics of primary and secondary outcomes by treatment and time have been estimated. Subgroup analysis was also employed according to baseline characteristics to explore whether the estimates vary by characteristics, and age (continuous variable) was cut off by median value (results given in Table 3, Table 4 and Table 5, respectively). Sensitive analysis was performed based on covariate analysis. All analyses were performed on intention to treat basis using the Statistical analysis software SAS 9.4. by SAS Institute, Cary, NC, USA. Figure 2 is produced using R software (version 4.1.0, http://www.R-project.org accessed on 1 January 2022).

## 3. Results

### 3.1. Training Competency in the Thinking Healthy Programme

Mean (M) and standard deviation (SD) of ENACT scores measured immediately after training were 45.73 (4.03) and 47.08 (4.53) for the e-training and face-to-face training groups, respectively. GLMM analysis showed a mean difference (MD) of −1.35, 95% CI −3.17, 0.46, and *p* < 0.001. Since 0.46 is much lower than 5 (non-inferiority margin), non-inferiority of e-training to face-to-face training can be established in terms of the primary outcome. At three months, the e-training arm mean scores (M) were 42.16 with an SD of 4.85, while the face-to-face group were M = 42.65 and SD 4.65. The mean difference (MD) between the two groups was −0.48, 95% CI −2.35, 1.39, and *p* = 0.61. The competence levels in terms of percentage of students achieving ≥80% competency was 41/49 (83.7%) students in the e-training arm and 42/47 (89.4%) in the face-to-face arm; the intervention method was not associated with competence (OR = 0.61, 95% CI: 0.17, 2.10, *p* = 0.42) (Table 5). These outcomes were partially sustained at three months where 23/44 (53.3%) students in the e-training arm and 21/47 (44.7%) students in the face-to-face arm achieved ≥80% scores, and these differences were not associated with the training method (OR = 1.36, 95% CI: 0.57, 3.26, *p* = 0.47). The lasting effects for intervention and control arms are −3.57 (45.73 in post-training, 42.16 in post three-months training) and −4.43 (47.08 in post-training, 42.65 in post three-months training. The findings from the covariate analysis and imputation analysis were similar, which demonstrates the robustness of the analyses (Table 2). The individual data plots of ENACT scores of the intervention and control groups from post-training to post three months is shown in Figure 2.

### 3.2. Attitudes and Beliefs about Perinatal Depression

Higher scores indicate the presence of more negative attitudes about perinatal depression. Moreover, the attitudes and beliefs scale yielded an adequate internal consistency with alpha coefficient estimated at 0.75. There was no significant difference in the scores of the two arms pre- and post-training [e-training M = 32.81, SD = 8.60 vs. face-to-face M = 30.66, SD = 7.38, mean difference (MD) = 2.15, 95% CI −1.33, 5.63, *p* = 0.22] and at three months post-training [M = 30.78, SD = 7.72 vs. M = 33.47, SD = 10.01, MD = −1.68, 95% CI −5.35, 1.98, *p* = 0.36]. The opposite lasting effects were seen for the two arms.

### 3.3. Self-Efficacy

There was no significant difference in self-efficacy scores of the students in both arms immediately post-training [e-training group M = 178.20, SD = 54.19 vs. face-to-face M = 187.24, SD = 47.05, mean difference (MD) = −19.13, 95% CI −39.54, 1.27, *p* = 0.06]. At three months post-training, the scores were e-training mean = 159.11, SD = 53.88 vs. face-to-face M = 178.24, SD = 40.78, mean difference (MD) = −9.17, 95% CI −30.44, 12.08, *p* = 0.39]. The Cronbach alpha values for the self-efficacy scale were part 1 (0.96), part 2 (0.94), and part 3 (0.97). These values were above the cut-off values of 0.90 (excellent internal consistency), but values higher than 0.95 can indicate some similar constructs and did not contribute uniqueness and may be redundant. However, in the present case, the concept of self-efficacy is a very narrow construct, and therefore, contained more homogenous items. Besides this homogeneity, it also had a high number of items, and both of these reasons may have inflated the alpha coefficients, as the Cronbach alpha is a function of item homogeneity and numbers, where more homogenous and a larger number of items yield a higher alpha coefficient [34].

### 3.4. Satisfaction with Training

When asked how much they had learnt from the course, 78.5% of participants in the e-training reported learning a great deal, compared to 75.6% in the face-to-face group. Similarly, 72.5% of participants in the e-training group reported finding it very useful, compared to 67.4% in the face-to-face group. When asked if they would use the training material in the future, 89.1% of those in the e-training group and 89.0% of those in the face-to-face group replied in the affirmative. When asked if they would recommend the course to others, 72.3% of those in the e-training group and 83.5% of those in the face-to-face group reported that they definitely would.

## 4. Discussion

The study found that technology assisted e-training is not inferior to specialist-led face-to-face classroom training in psychosocial management of perinatal depression in student nurses in Xi’an, China. The results showed that similar levels of competency were achieved in both groups immediately post-training and at three-months follow-up. The lasting effects for intervention and control arms show that the effectiveness of the control arm could last for a shorter time than that of intervention arm. Self-reported efficacy levels and satisfaction with training were also similar for both forms of training. This study contributes to the very scarce literature on the modes of training for delivery of psychosocial interventions in low- and middle-income countries, especially in settings where specialist trainers and training centres are few and concentrated in urban centres.

While studies exist on on-line or remote training methods in mental health, very few have evaluated such training in evidence-based psychological and psychosocial interventions for non-specialists, such as nurses and allied health professionals [35,36]. Very few on-line or technology-assisted programmes focusing on non-specialists have been evaluated globally. A recent systematic review evaluated 28 randomized controlled studies comparing the effectiveness of evidence-based psychotherapies training modalities on provider and client outcomes [26]. Training modalities included face-to-face workshops, online training, and self-study using instructional manuals. Most of the studies measured provider outcomes (such as competence, satisfaction, and fidelity). No major differences were found between online and face-to-face modalities. However, all these studies were from high-income countries. Our previous work in Pakistan [19] conducted using a similar methodology to the current study showed very similar results, and the findings of both studies are in line with the research from high income countries [26]. Innovative aspects of our study are that it is was conducted on student nurses rather than experienced health professionals, integrated into their curriculum, and delivered through routine training systems. The successful uptake in this context has implications for training of not only nurses but other health professionals in evidence-based psychological therapies.

We did not evaluate implementation outcomes, such as adoption of the intervention into practice after training, additional costs of training, or service level outcomes, such as impact of the training on patient care. Another potential weakness is the limited study settings (only two institutions in one city). Further long-term research is needed to evaluate the impact of scaled-up training on patient care in multiple sites. However, this study clearly demonstrated that nursing students developed new skills, became competent in using them, and felt confident in delivering a new intervention. Such training can lay the foundations for integrated systems of continued professional development and familiarise health professionals to psychosocial interventions at an early stage in their careers.

Our study was limited to the evaluation of classroom training in nursing students. These students did not go on to practice their skills under supervision, as would normally be the case. We found that there was a drop in the competency levels of the students after three months. This finding underlines the importance of ongoing practice under supervision for the maintenance of new skills acquired from the training. Many models of supervision appropriate to our settings have been described. Notable amongst these are the apprenticeship [37] and cascaded models [38]. Both models, developed for low- and middle-income settings have minimal dependence on specialists and instead rely on peer-learning, sharing of experiences, and problem solving in a facilitated environment, with only the most complex issues referred for specialist input. Future research would be required to establish and evaluate such supervision systems in Chinese nursing educational and service systems.

Our study has important public health implications. Previously regarded as the domain of specialists, the area of psychosocial intervention in recent decades has opened up to practice by non-specialists [39]. Many studies demonstrate that non-specialists can provide evidence-based psychological interventions in the area of perinatal depression [14], and this has led to recommendations by the WHO to include the Thinking Healthy Programme in its mental health gap action programme and recommend its delivery by non-specialist health workers. However, a bottle-neck to scale up the programme remains due to the scarcity of specialist trainers to provide training to hundreds of thousands of non-specialist health professionals, such as community health workers, nurses, and midwives [13]. The number of mental health professionals has increased in China in recent years; however, these resources are mostly located in urban psychiatric hospitals, making services far less accessible for at least half of China’s 1.49 billion people living in rural areas. The Chinese government is constantly investing in increasing the number of psychiatrists in the coming years and promoting digital approaches in mental health care. Online training of existing nurses for delivering psychosocial therapies can be a step to attain the required human resource for mental health coverage. However, this training must be accompanied by regular practice under supervision. As, telemedicine and e-health have gone viral globally and sometimes looks like a silver bullet post COVID-19, digital solutions, such as telemedicine and online health monitoring, are quickly shifting from a previously slow adoption path to a quick pace of acceptance [40]. In such critical times, e-trainings for evidence-based psychological interventions can be a way forward in both high- and low-income settings.

## 5. Conclusions

Our study shows that both e-training and face-to-face specialist-led training modalities are equally effective in achieving competency, fidelity, and self-efficacy for delivering evidence-based psychosocial interventions and improving knowledge, attitudes, and counselling skills. The relative ease with which this form of training was integrated into existing educational systems and the satisfaction of students with it, opens promising avenues to build an effective mental health-nursing workforce in China, which can be a model for other countries in the world. e-training can be used for training of non-specialists for evidence-based psychosocial interventions at scale and can be utilized where there is a shortage of specialist trainers, but practice under supervision is necessary to maintain competence.

## Figures and Tables

**Figure 1 ijerph-19-04094-f001:**
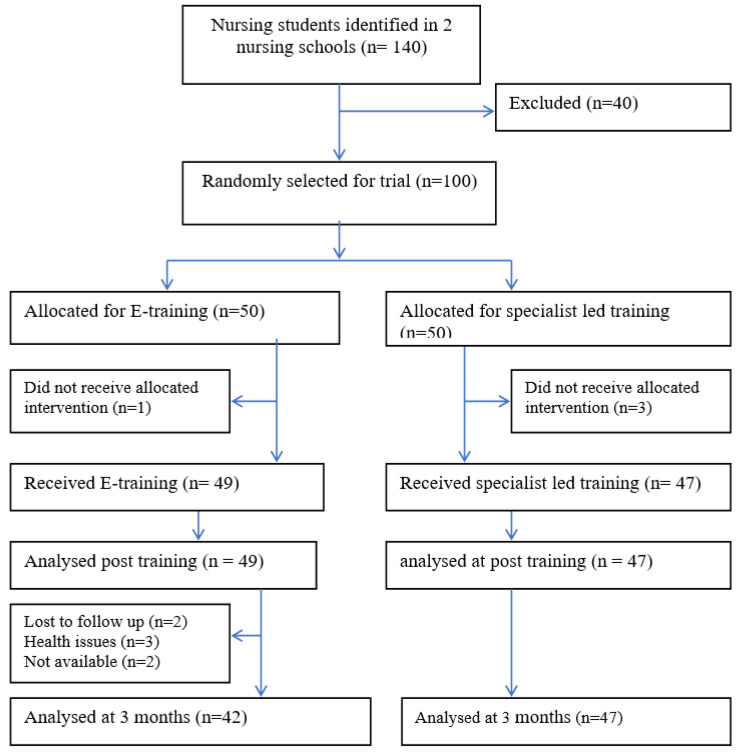
(CONSORT flow diagram).

**Figure 2 ijerph-19-04094-f002:**
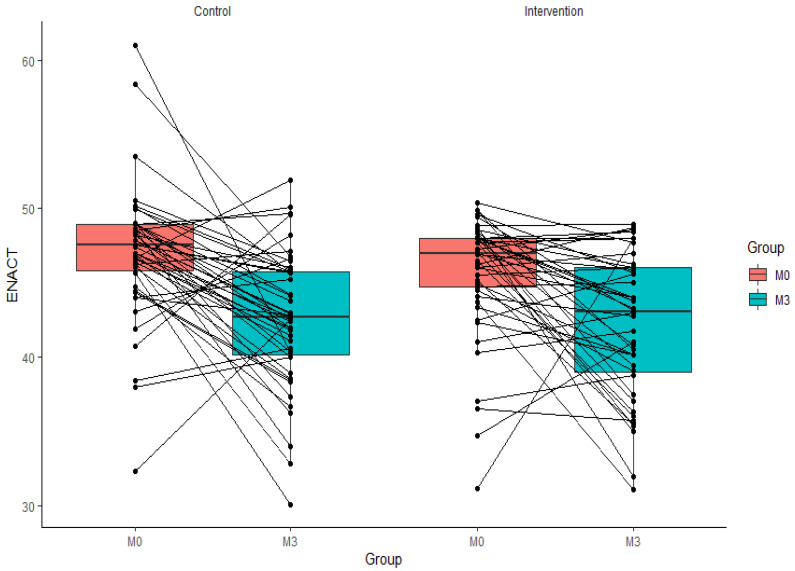
Scatter plot of ENACT scores of the intervention and control groups from post-training to post three months.

**Table 1 ijerph-19-04094-t001:** Socio demographic characteristics of intervention and control arms.

Characteristic	E-Training (*n* = 50)	Specialist Led Training (*n* = 50)	*p*-Value
Mean age (SD), Median (IQR)	20 (1.94); 20 (18–22)	19.45 (1.57); 19 (18–21)	0.13
Gender			0.08
Male	7 (14.9%)	14 (29.8%)	
Female	40 (85.1%)	33 (70.2%)	
Grade			0.09
Year 1	23 (48.9%)	31 (66.0%)	
Year 4	24 (51.1%)	16 (34.0%)	
Work experience (*n* [%]) *			1.00
Yes	10 (21.3%)	10 (21.3%)	
No	37 (78.7%)	37 (78.7%)	
Prior mental health training (*n* [%]) **	10.12	11.02	0.61
Yes	3 (6.4%)	1 (2.1%)	
No	44 (93.6%)	46 (97.9%)	
Knowledge of perinatal depression **	17.04 (1.40)	17.23 (1.13)	0.43

* Compared using chi-square test. ** Compared using Fisher exact test. IQR, interquartile range.

**Table 2 ijerph-19-04094-t002:** Mean differences in primary and secondary outcome scores (competence) at post-training and three months post-training.

Generalized Linear Mixed Model Analysis *						
	Primary Analysis		Covariate Analysis		Imputation Analysis	
	Mean Difference (95% CI)	*p*-Value	Mean Difference (95% CI)	*p*-Value	Mean Difference (95% CI)	*p*-Value
ENACT Scores						
Post-training	−1.35 (−3.17, 0.46)	0.14	−1.20 (−3.05, 0.65)	0.20	−1.18 (−3.04, 0.66)	0.20
Post 3 months	−0.48 (−2.35, 1.39)	0.61	−0.32 (−2.23, 1.59)	0.73	0.16 (−1.69, 2.01)	0.86
Attitude and beliefs scores						
Post-training	2.149 (−1.338, 5.636)	0.22	0.49 (0.01, 0.98)	0.16	0.48 (0.05, 0.92)	0.28
Post 3 months	−1.686 (−5.358, 1.986)	0.36	−1.544 (−5.275, 2.188)	0.41	−1.532 (−5.080, 2.015)	0.39
Self-Efficacy scores						
Post-training	−19.13 (39.54, 1.27)	0.06	−16.44 (−37.23, 4.34)	0.11	−15.93 (−35.71, 3.84)	0.11
Post 3 months	−9.17 (−30.44, 12.08)	0.39	−6.52 (−28.22, 15.16)	0.55	−7.24 (−27.02, 12.53)	0.46

* GLMM: Generalised linear mixed models (GLMM). GLMM has study arm (intervention or control), follow-up visit (Post training or 3-months follow up), interaction of study arm and follow-up visit as fixed effects, and subject as random effect * *p* value for non-inferiority test.

**Table 3 ijerph-19-04094-t003:** Summary statistics of primary and secondary outcomes by treatment and time.

Category	Outcome	Month	Statistics	E-Training	Specialist Led Training	All
Primary outcome	ENACT	Post training	*n*, mean (SD)	49, 45.73 (4.03)	47, 47.08 (4.53)	96, 46.39 (4.31)
		Post 3 months	*n*, mean (SD)	44, 42.16 (4.85)	47, 42.65 (4.56)	91, 42.41 (4.68)
Secondary outcome	Attitude and Beliefs	Post training	*n*, mean (SD)	47, 32.81 (8.60)	47, 30.66 (7.38)	94, 31.73 (8.04)
		Post 3 months	*n*, mean (SD)	40, 31.78 (7.72)	45, 33.47 (10.01)	85, 32.67 (9.00)
	Self-efficacy	Post training	*n*, mean (SD)	47, 159.11 (53.88)	45, 178.24 (40.78)	92, 168.47 (48.62)
		Post 3 months	*n*, mean (SD)	40, 178.20 (54.19)	45, 187.38 (47.05)	85, 183.06 (50.44)

SD, standard deviation.

**Table 4 ijerph-19-04094-t004:** Estimates of treatment differences in means of primary outcome: Subgroup analysis *.

Variable	Subgroup	Mean Difference	*n*	Estimate (95% CI)	*p*-Value
Age	<19	A vs. B at month 0	52	−1.19 (−3.83, 1.44)	0.36
		A vs. B at month 3	52	−0.86 (−3.63, 1.89)	0.53
	≥19	A vs. B at month 0	44	−1.51 (−4.13, 1.09)	0.24
		A vs. B at month 3	44	−0.28 (−2.92, 2.39)	0.82
Gender	Male	A vs. B at month 0	21	−1.26 (−5.47, 2.95)	0.53
		A vs. B at month 3	21	−0.51 (−4.72, 3.69)	0.80
	Female	A vs. B at month 0	75	−1.18 (−3.31, 0.94)	0.26
		A vs. B at month 3	75	−0.41 (−2.60, 1.77)	0.70
Grade	1st year	A vs. B at month 0	56	−1.04 (−3.52, 1.43)	0.40
		A vs. B at month 3	56	−0.35 (−3.00, 2.28)	0.78
	4th year	A vs. B at month 0	40	−1.66 (−4.58, 1.25)	0.25
		A vs. B at month 3	40	−0.76 (−3.67, 2.15)	0.59
Work experience	Yes	A vs. B at month 0	20	−1.34 (−4.53, 1.85)	0.38
		A vs. B at month 3	20	−1.08 (−4.36, 2.18)	0.49
	No	A vs. B at month 0	76	−1.34 (−3.51, 0.82)	0.22
		A vs. B at month 3	76	−0.30 (−2.53, 1.92)	0.78

* Not adjusted for covariates. A = e-training, B = specialist led training.

**Table 5 ijerph-19-04094-t005:** Estimates of odds ratio of primary outcomes from generalized linear mixed models.

		Primary Analysis *		Covariate Analysis **		Imputation Analysis **^†^	
Outcome	Odds Ratio	Estimate (95% CI)	*p*-Value	Estimate (95% CI)	*p*-Value	Estimate (95% CI)	*p*-Value
ENACT (≥43)	A vs. B at month 0	0.61 (0.17, 2.10)	0.42	0.59 (0.16, 2.10)	0.41	0.59 (0.16, 2.12)	0.42
	A vs. B at month 3	1.36 (0.57, 3.26)	0.47	1.33 (0.53, 3.37)	0.53	1.67 (0.67, 4.15)	0.26

* Not adjusted for covariates. ** Adjusted for age, gender, grade, work experience and prior mental health training where missing data were imputed with medians. † Missing outcome data were imputed using the last observation carried forward method. A = e-training, B = specialist led training.

## Data Availability

Not applicable.

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
