# Peer review of "Standardising Training of Nurses in an Evidence-Based Psychosocial Intervention for Perinatal Depression: Randomized Trial of Electronic vs. Face-to-Face Training in China"

_ijerph, 2022, doi:10.3390/ijerph19074094_

Round 1
Reviewer 1 Report
Thank you for the opportunity to review this manuscript. The time spent on its creation and submission is greatly appreciated. The topic they address is very interesting and novel. Perhaps the most relevant aspect of the work is that important practical applications emerge from it that can be taken into account and applied by the health authorities. However, there are some issues that I suggest you be aware of and fix. I present them below, hoping they will be useful. -It would be necessary to include in the introduction the mention of health training programs implemented online and with satisfactory results, in addition to the one used in Pakistan. Mention the problems they have had, how the adaptation has been, if they are currently in use... Once you have completed the introduction with this type of information, you can state the hypotheses of the work. -In the case of validated instruments, include the confidence results. On the other hand, in the case of ad hoc instruments, it is recommended to include exploratory factor analyzes to check the validity of the tests. -Sections 3.1. 3.2. They are part of the work methodology. Keep in mind that the results section aims to respond to the objectives of your work. And among them is not the description of the sample or the selection of the participants. Therefore, information 3.1 and 3.2 must be restructured to be included in the methodology. -The tables, especially 2, have a very messy presentation that makes it difficult to see and understand. Please check this. -The conclusions are very vague. Expand this section. Thank you for your work.
Author Response
Dear Reviewer,
We deeply appreciate your kind help with the above manuscript and have revised the manuscript accordingly. We have found the comments tremendously helpful. In the revised version of our manuscript together with our point-by-point replies, you will see that the necessary amendments have been made based on the kind suggestions.
We sincerely hope that the updated version is now acceptable. Should you feel that there are other changes required, please do let us know.
Regards,
Xiaomei Li

Reviewer 2 Report
In this manuscript, the authors investigated the effectiveness of an e-training program of psychological intervention for perinatal depression in China as compared to face-to-face training. The study is generally well conducted and the manuscript clearly written. The reviewer, however, has several concerns for the authors to address, which may help improve the manuscript.
1), abstract, the meaning of ENACT is unclear and should be explained. Based on the results reported in lines 21-22, it is hard to reach the conclusion that the e-training is not inferior, given that the difference is significant at p<0.01, more explanation should be given to avoid confusion.
2), the 1st sentence of the introduction needs references.
3), for the measures or scales used in this study, data of internal consistency should be reported to validate that the scales can be used in this context.
4), the authors should give a brief description of missing data. Furthermore, rather than the last observatio ncarried forward, multiple imputation may be preferable.
5), the 1st sentence of section 3.2, please also report the sd of age. Table 1, "mean age n, median (IQR)" may have to be revised.
6), for data visualization purpose, can the authors provide a scatterplot showing the individual data plots of ENACT scores of the intervention and control groups from post training to post 3 months?
Author Response
Dear Reviewer,
We deeply appreciate your kind help with the above manuscript and have revised the manuscript accordingly. We have found the comments tremendously helpful. In the revised version of our manuscript together with our point-by-point replies, you will see that the necessary amendments have been made based on the kind suggestions.
We sincerely hope that the updated version is now acceptable. Should you feel that there are other changes required, please do let us know.
Best regards,
Xiaomei Li

Reviewer 3 Report
It is a timely and important study for developing countries, which are always lacking health-related professionals to build up their health care systems. The training programs for non-specialists become very important to developing countries. The research design is well-designed to examine the effectiveness of the training program. However, I would like to raise some minor concerns for the paper as follows:
- Cultural adaptation of the programme. As the authors propose to make use of their Pakistan study and they simply convert the programme by the Mandarin-speaking authors, the authors should discuss about the cultural adaptation of the programme, i.e. back-translation procedure, or the similarity of Thinking Healthy training manual between Pakistan and China.
- Comparison between intervention arm and control arm. As the descriptions between the arms are quite different, I cannot find out the difference between the two arms easily. I propose the authors write up a table to summarize the difference of the two arms directly.
- The integrity of the scales. The authors may not make use of the complete set of some measurement tools proposed in the study. For example, the authors simply delete adoption from provider-level outcomes without explanation. Similarly, they simply delete item 17 and item 18 from ENACT. Moreover, the authors propose a cutoff point of ENACT, I cannot understand the validity of modified ENACT score with “the cut-off score of 80% or more to indicate competency” (Finally the authors make use of ENACT score rather than the competency derived from the proposed cut-off score). It is better for the authors to explain the arrangements.
- Input of missing outcome data. The authors may need to explain the reason for missing outcome data inputted by the last observation carried forward method. As we know, the effectiveness of training on the trainees is always decaying after training. Therefore, the results of 3-months post-training may be affected.
- Additional findings from table 2. The two arms may have different lasting effects. For the perspective of “Attitude and beliefs” in table 2, the lasting effects for intervention and control arms are -1.03 (32.81 in post training, 31.78 in post 3-months training) and 2.81 (30.66 in post training, 33.47 in post 3-months training), so we can see that there are opposite lasting effects for the two arms. On the other hand, for the perspective of “ENACT Scores”, the lasting effects for intervention and control arms are -3.57 (45.73 in post training, 42.16 in post 3-months training) and -4.43 (47.08 in post training, 42.65 in post 3-months training), so we can see that the effectiveness of the control arm could last for shorter time than that of intervention arm. Maybe it is worthwhile for the authors to discuss about the lasting effects of the arms.
- Readers may not be very familiar with Thinking Healthy Programme. The authors are highly recommended to introduce Thinking Healthy Programme, especially the Chinese version.
Author Response

(The authors gave the same response as above.)

Round 2
Reviewer 2 Report
6. for data visualization purpose, can the authors provide a scatterplot showing the individual data plots of ENACT scores of the intervention and control groups from post training to post 3 months?
Response: Thank you very much for the suggestion. Scatter plot has been added on page 10 of manuscript, as shown below.
Thank the authors for providing the figure. A smaller change that may greatly enhance the figure is to show the within-subject change from postintervention to 3 months later. A figure like Fig. 3 of the following figure is better: https://www.sciencedirect.com/science/article/pii/S0095447017301407#f0020
More example:
https://stackoverflow.com/questions/57763876/how-to-combine-geom-bar-geom-point-and-geom-line-into-one-figure
https://stackoverflow.com/questions/52743835/creating-a-multiple-dot-plot-box-plot-line-plot-with-ggplot2
If the figure looks too complicated, the authors may plot intervention and control groups separately on two subfigures. Hope this may help.
